# Managerial Issues Regarding the Role of Natural Gas in the Transition of Energy and the Impact of Natural Gas Consumption on the GDP of Selected Countries

**Eleftherios Thalassinos** [1,2], **Marta Kadłubek** [3,*], **Le Minh Thong** [4], **Tran Van Hiep** [4] **and Erginbay Ugurlu** [5]

1   Faculty of Maritime and Industrial Studies, University of Piraeus, 185-33 Piraeus, Greece; thalassinos@ersj.eu
2   Faculty of Economics, Management and Accountancy, University of Malta, 2080 Msida, Malta
3   Faculty of Management, Czestochowa University of Technology, 42-200 Czestochowa, Poland
4   Faculty of Economics and Business Administration, Hanoi University of Mining and Geology, Hà Nội 100000, Vietnam; leminhthong@humg.edu.vn (L.M.T.); tranvanhiep@humg.edu.vn (T.V.H.)
5   Department of International Trade and Finance, Istanbul Aydin University, 34295 Istanbul, Turkey; erginbayugurlu@aydin.edu.tr
*   Correspondence: martakadlubek@wp.pl

**Abstract:** Natural gas is considered an important bridge in the transition of energy in the world. However, the development and management of unconventional gas now face many challenges. This paper aims to provide an overview of what would be required to formulate and implement the trend of energy transition and natural gas use in the world. Selected managerial issues regarding the role of natural gas are presented, including chosen statistical data on natural gas consumption, forecasts for the demand for natural gas, and the potential of unconventional gas. The empirical part of the study examines the impact of natural gas consumption on the GDP of 14 G20 countries during the period of 1994 to 2018. The pooled mean group model (PMG) is employed in this study. Based on the cointegration test results, it was found that natural gas consumption and population have a long-run relationship with $CO_2$ emissions. Consistent with other studies, we found a positive relationship between $CO_2$ emissions and natural gas consumption, GDP, and population.

**Keywords:** natural gas; unconventional gas; shale gas; energy transition; energy management model

## 1. Introduction

The growing pace of life in the 21st century is the reason for the greater demand for electricity and fuel. The raw materials used in conventional energy are limited in resources and not equally available to all; moreover, their combustion results in the emission of harmful substances. The solution to this problem is transformations in the management [1–4] of this area towards sustainable development of alternative sources. In this context, sustainable development is understood as the efficient use of energy resources to meet the needs of current and future generations for electricity and fuels in a manner that does not disturb the ecological balance. It means that economic development must be treated synonymously with the protection of the environment. Therefore, the transformations in the management of this area should be focused on the choice of energy-efficient and ecological solutions. This balance can be achieved thanks to advances in the use of renewable energy sources, including the use of biological methods of fuel and energy production. The benefits of using them in the energy economy are energy security, market competitiveness, improvement of the quality of life, reduction of the amount of waste, and emission of pollutants into the atmosphere.

The World Energy Council [5] estimates that the global hard coal resources will last for about 150 years at the current production level. The search for alternative energy sources is carried out all over the world. The global economy will shift from hard coal-based energy

to other energy sources at a different pace between countries, depending on their overall level of development and financial capacity and advances in energy technology.

Experts in the theory of socioeconomic development believe that of the six key trends determining the global situation in the future, the energy factor occupies a leading position. This is confirmed by the history of our civilization's development to date. The driving force of progress was the increasing consumption of various energy sources and their relatively easy availability. Moreover, they should be relatively cheap. However, competition for access to them has caused a series of sharp diplomatic tensions, including the occurrence of armed conflicts. On the other hand, one should not forget about the scarcity of natural resources, especially fossil fuels, the primitive combustion of which results in adverse climate change.

Sustainable development, recognized as the constitutional principle of many countries in the world, is defined as socioeconomic development in which the process of integrating political, economic, and social activities takes place while maintaining the natural balance and durability of basic natural processes to guarantee the possibility of satisfying the basic needs of individual communities or citizens of both the present and future generations [6]. There is no separate strategy for sustainable development; however, many strategic documents define long-term socio-economic goals and activities in line with the principle of sustainable development, considering social, economic, and ecological cohesion [7]. Energy security undoubtedly belongs to the context of sustainable development. This concept has been defined in many legal acts and is still being redefined. In general, it can be assumed that energy security is the state of the economy that allows for the covering of the current and future demand of consumers for fuels and energy in a technically and economically justified manner while minimizing the negative impact of the energy sector on the environment and living conditions of the society [8].

The road to sustainable development is an opportunity to increase innovation and productivity in the global economy while improving the social and environmental situation on a global scale. However, the current approach must be changed from focusing on maximizing profit to striving to create common values [9]. Creating common values should affect the transformation in the management of structures, processes, and systems, and requires going beyond the framework applicable in each industry and the ability to predict upcoming challenges and opportunities related to potential threats [10]. Strategic national decisions should undertake to transform the foundations on which they are built by introducing an energy-friendly and sustainable development model, which will increase its competitiveness in the long term while also improving the material, social and environmental conditions of the community [11].

For many years, ensuring energy security has been a serious dilemma for many countries. This issue is particularly relevant to Europe, which is poor in strategic raw materials. The insufficient resources of energy-intensive economies require a steady and reliable supply of the necessary fossil fuels. In this respect, natural gas is undoubtedly one of the most important. Blue fuel differs from coal or crude oil in that its combustion produces far fewer chemicals that pollute the natural environment. There are also other factors in favor of the increasing use of natural gas—extraction, storage and transport are carried out in more environmentally friendly conditions than in the case of other fuels. Unfortunately, many countries are forced to import gas due to limited gas resources. Additionally, significant dependence on a primary supplier is a disadvantage. It should be noted that the development of the global gas sector could significantly modify the structure of electricity generation, which is largely based on coal.

The world is undergoing a radical transition to a low-carbon economy, reducing dependence on fossil fuels and adapting to climate change. Low carbon energy is an energy source that produces less greenhouse gas emissions than traditional energy sources such as wind, solar, geothermal and nuclear. In addition, low-carbon energy also includes low-emission energy sources such as natural gas and processing technology development to capture carbon.

However, compared with traditional energy sources, developing these types of energy and technologies requires a huge investment in technology, investment capital, energy managerial models and time. With renewable energy, the most important issues right now are technology and cost. As for nuclear energy, although it is a clean energy source, its safety is still controversial.

In the present context, natural gas is considered a bridging energy source in the process of human transition from traditional energy sources to renewable energy due to its superior properties. Although natural gas is also a fossil fuel source, natural gas is a cleaner energy source than oil and coal. Table A1 in Appendix A shows the superiority of natural gas compared to other traditional fossil fuel sources [12].

When the same amount is burned, natural gas emits minimal $CO_2$, only half that of coal, and 75% of oil. It also emits tiny particles of other toxins and produces dust as well as mercury. Thus, natural gas is considered an environmentally friendly fuel source widely used in many fields. It will continue to be used more than other energy fossils in the future. According to the world energy organization's forecast, humanity is entering the golden age of natural gas to replace the era of oil.

Relative to the concern of this paper, natural gas, and sustainable development, which is criticized above, we aim to identify the impact of natural gas consumption on $CO_2$ emissions. We aim to recognize a general review of the world, and to represent the world; we use a panel of G20 countries. G20 countries are the highest $CO_2$ emitter worldwide and the highest energy consumer country group; we investigate the long-run equilibrium relationship among the variables as well as their short-run relationship. In this study, the Pooled Mean Group (PMG) model of panel Autoregressive Distributed Lags (ARDL) will be employed for the period from 1994 to 2018. The PMG model is one of the panel data econometric techniques, and both PMG and panel data have many advantages. Firstly, PMG allows us to test the long-run relationship between variables, and the panel data reduces the effects of collinearity among the explanatory variables and provides more degrees of freedom. Moreover, the panel data increases efficiency and reduce the problems arising from substandard distributions. Furthermore, by using the PMG/ARDL model and the error correction model, we can see the short-run adjustments of each model.

The paper makes several contributions to energy and ecology literature. This is a unique paper investigating the effect of natural gas consumption on $CO_2$ emission for selected G20 countries from 1994 to 2018. In addition, the paper uses a PMG/ARDL model with the data, which shows both long-run and short-run relationships. Finally, this paper shows the results of a panel of G20 countries and their cross-sectional results.

In this paper, we elaborate on the countries, which are the world's 20 largest economies (G20). Because more than 80% of global energy consumption is caused by G20 countries and, they have the largest $CO_2$ emissions in the world with huge coal (95%), oil and gas (70%) consumption share within the other countries [13]. The G20 consists of Argentina, Australia, Brazil, Canada, China, France, Germany, India, Indonesia, Italy, Japan, South Korea, Mexico, Russia, Saudi Arabia, South Africa, Turkey, the United Kingdom, the United States and the European Union.

The paper is organized as follows. In Section 2, the literature review is presented. In Section 3, the methodology of the research is outlined. Section 4 presents the results of the research; Section 5 discusses the results, and Section 6 concludes the paper.

## 2. Literature Review

### 2.1. Natural Gas Consumption

Although natural gas has been known for a long time, its exploitation and widespread use developed in the early twentieth century due to scientific and technological development. According to BP data, the demand for natural gas around the world has increased very rapidly in recent decades. Natural gas is the world's third-most used energy source after oil and coal [14]. According to BP statistics in 2019, over the past five decades, natural gas consumption has increased almost fourfold from 891 Mtoe in 1970 to 2209 Mtoe in 2018.

Its share of total energy consumption global growth increased from 18% in 1970 to 25% in 2018 [15].

Natural gas plays an increasingly important role in many economic sectors. Its use has extended to most sectors of energy consumption. The main industries using natural gas are electricity, residential, industry, and transportation, while the electricity generation sector accounts for the most significant proportion of the distribution of natural gas by fields of use [15].

The consumption of natural gas in the world has increased year by year. In particular, the growth of global natural gas demand after 2000 mainly came from Asian countries (mainly China, India), the Middle East and the recovery of demand for natural gas in the United States from 2007. According to BP data in 2019, the average growth rate of natural gas consumption in the world between 2007 and 2017 was 2.2% per year. The Middle East and Asia-Pacific region have the highest growth rate, with the corresponding rate of 5.6% and 5.0% per year. In 1980, gas consumption was mainly concentrated in North America and Europe, with nearly 90% of the total world output. By 2018, these two regions' total consumption volume only accounts for nearly 41% of the total worldwide consumption; the Middle East and Asia-Pacific region increasingly consume about 36% of the total world consumption [15].

### 2.2. Natural Gas Reserves

According to estimates of recent studies, global natural gas reserves are plentiful and progressive due to technological development. Because of improved exploration methods, the world's natural gas reserves are increasing. In particular, the rapid development of recent technology has allowed for the exploitation of unconventional gases that are considered to have very large reserves, most notably the recent shale oil and gas revolution in the United States. The discovery of non-traditional natural gas fields has changed the natural gas reserve picture and has affected geopolitics in many regions of the world. For example, according to [16], in the US, shale gas reserves account for more than four times the reserves of conventional gas, which greatly impacts the US's future energy development strategy.

At the end of 2018, according to [15] statistics, proven natural gas reserves were about 197 $Tm^3$ and equivalent to over 51 years of consumption at current levels. The increase in proven natural gas reserves over the years has been much faster than gas production in some countries. The average annual growth rate of the world's natural gas reserves over the last ten years is 1.9% per year. From 2007 to now, North America has the highest natural gas reserve growth rate globally, with an average growth rate of 5.3%. Mainly contributing to this increase in reserves is the development of non-traditional gases, especially the shale gas revolution in the last decade in the United States. In addition, there is an increase from the area of the former Soviet Union countries and the Asia-Pacific region with an average growth rate of 4.4% and 3.0% per year, respectively. Natural gas reserves are still concentrated mainly in the Middle East, where huge reserves account for 38.4% of world reserves, followed by regions of the former Soviet Union with 31.9% [15]. The natural gas reserves in the world are still on an increasing trend. In the future, traditional natural gas may still be found in unknown areas or recovered from known sedimentary basins. Furthermore, there is also the development trend of non-traditional gas in many countries and regions around the world.

### 2.3. Forecasts for the Demand for Natural Gas

The growth of the global economy and population growth leads to an increase in energy demand and consumption. However, it is forecasted that energy consumption growth will begin to slow down after 2040 compared to the recent period. According to the International Energy Agency's 2018 energy outlook report, the growth rate of world demand from 2017 to 2040 in the New Policy scenario is about 1.1% per year [17].

According to forecast scenarios [18–26], the world's energy demand is likely to increase by 40% between now and 2040. Much of the most significant energy demand increase will come from developing countries (non-OECD). Developing countries in Asia and the Middle East will account for three-quarters of the increase in global demand by 2040. In Asia, China and India are the two countries with the largest energy demand growth rates globally. India's energy demand up to 2040 will be twice that of the current level and approximately half of China's demand. Other regions of the world, such as the Middle East and Africa, also have very high demand growth—demand by 2040 will be 60% greater than now [17]. According to scientists, energy consumption is the largest cause of climate change, with about two-thirds of all human-made greenhouse gas emissions [27]. Therefore, there is a need to establish a sustainable and environmentally friendly energy system. This is a priority for energy and climate policymakers worldwide, with natural gas being an important bridge in the energy transition.

Consequently, according to many estimates, the demand for natural gas is expected to increase more than any other fossil energy source. All energy scenarios of energy organizations in the world, such as the International Energy Agency (IEA), World Energy Council (WEC), or oil companies such as Shell, ExxonMobil, and BP, offer a promising long-term future gas. In many scenarios, natural gas will be the world's leading energy source by 2050 [28]. For example, according to ExxonMobil's analysis, 40% of global energy demand growth between 2014 and 2020 is expected to be met by natural gas [29].

Similarly, according to the IEA in the report, "Are We Entering a Golden Age of Gas?", due to more natural gas consumption, the world could achieve the overall goal of reducing $CO_2$ emissions [20]. According to the IEA, global natural gas demand is expected to grow 50% between 2014 and 2040, which is faster than other fuels and twice as fast as oil. Most of the increase in natural gas demand comes from emerging economies, with China and India accounting for about 30% of the increase and the Middle East at more than 20% [20–24].

According to the IEA and the scenarios in their "World Energy Outlook" reports from 2010 to 2018 [17–28], demand for natural gas will steadily increase. Still, the rate of increase varies from year to year and from region to region. Table 1 shows the growth rate of natural gas under the IEA scenario.

**Table 1.** Growth rates of natural gas in New Policies Scenarios of International Energy Agency (%).

| Region | WEO 2010 | WEO 2011 | WEO 2012 | WEO 2013 | WEO 2014 | WEO 2015 | WEO 2016 | WEO 2017 | WEO 2018 |
|---|---|---|---|---|---|---|---|---|---|
| Total world energy | 1.20 | 1.3 | 1.2 | 1.2 | 1.1 | 1.0 | 1.0 | 1.0 | 1.0 |
| Petroleum demand | 0.5 | 0.6 | 0.5 | 0.5 | 0.5 | 0.4 | 0.4 | 0.5 | 0.5 |
| Coal demand | 0.6 | 0.8 | 0.8 | 0.7 | 0.5 | 0.4 | 0.2 | 0.2 | 0.1 |
| Gas demand | 1.4 | 1.7 | 1.6 | 1.6 | 1.6 | 1.4 | 1.5 | 1.6 | 1.6 |
| North America | 0.4 | 0.6 | 0.8 | 0.8 | 1.0 | 0.7 | 0.7 | 0.7 | 0.8 |
| Euro | 0.5 | 0.9 | 0.7 | 0.6 | 0.7 | 0.1 | 0.4 | 0.3 | −0.1 |
| Asia | 3.8 | 4.3 | 4.2 | 4.2 | 3.8 | 3.6 | 3.6 | 3.0 | 3.1 |

Source: [16–28].

The demand for natural gas increases faster than any other energy source. According to the [17–28] outlook forecasts for the last 10 years, the average increase in world demand for natural gas has ranged from 1.4% to 1.7% per year, while the largest increase for coal and oil is only 0.8% per year; even in recent forecasts, this growth rate tended to decrease sharply. According to the IEA's forecast, by 2040, natural gas will overtake coal as the second-largest source of energy in total primary energy demand. Around the world, the Asian region will be the main driver of growth in future natural gas demand, with a very high growth rate of 3.0% to 4.3% per year compared with 1.4% to 1.7% of the average worldwide growth rate (Table 2).

**Table 2.** Forecast the growth rate of GDP and population in the world and Asia.

| World | 2000 | 2005 | 2010 | 2015 | 2020 | 2025 | 2030 | 2040 | 2050 |
|---|---|---|---|---|---|---|---|---|---|
| The GDP growth rate (%) | | 4.53 | 5.03 | 3.80 | 4.09 | 3.75 | 3.43 | 2.85 | 2.34 |
| Population (M) | 6128 | 6514 | 6916 | 7325 | 7717 | 8083 | 8425 | 9039 | 9551 |
| **Asia** | **2000** | **2005** | **2010** | **2015** | **2020** | **2025** | **2030** | **2040** | **2050** |
| The GDP growth rate (%) | 0.0 | 8.64 | 9.36 | 6.32 | 6.50 | 5.45 | 4.59 | 3.53 | 2.47 |
| Population (M) | 3 287 | 3 483 | 3 670 | 3 858 | 4 026 | 4 168 | 4 285 | 4 441 | 4 496 |

Source: [30].

*2.4. The Need for a New Energy Management Model*

Unlike conventional gas, unconventional gas extraction is more complex and challenging due to its low permeability. Unconventional gas development is complex and multi-faceted, with economic, environmental, public health, social and technological components to consider. Unconventional gas exploitation projects often require large investment capital and different technologies. While the project life is short, production output declines rapidly. The development of unconventional oil and gas projects is vulnerable to market fluctuations, especially price factors.

Therefore, it is necessary to have an appropriate management model for developing unconventional gas sustainably, including all aspects related to its development, including finance and non-financial factors such as drilling, mine development, capital management, water resource management and use, and health and safety issues, etc.

In fact, the development of unconventional oil and gas companies has been facing many risks and difficulties in maintaining their development. From 2015 to the end of 2020, about 500 oil and gas companies have declared bankruptcy in North America, including the US oil and gas giant and a pioneer in shale oil exploitation (Chesapeake Energy), which filed for bankruptcy in June 2020 [31]. The collapse was that the companies did not have a suitable management model in the context of oversupply, leading to low energy prices, especially in the context of the significant impact of the COVID-19 pandemic. As the reaction to such a crisis, the advancement of sustainable solutions has confirmed their capabilities as an auspicious and useful strategy. To adequately consider the current consequences of the COVID-19 pandemic on renewable energy increase strategies, first, the short-term management concerns should be recognized. In contrast, the mid- and long-term approaches should be specified to attain precise renewable energy goals and proceed to a more socially and environmentally energy prospect [32]. Finally, despite the downturn, the energy sector is still one of the most significant areas in the world economy [33], which is undeniably an imperative determinant of searching for improvements within the management processes.

*2.5. Unconventional Gas Evolution and Its Effects*

Although unconventional gases have been known for a long time, the potential and development of non-traditional gases and their impact on the energy market are only about a decade ago. Today, known unconventional gases include coal-bed methane (CBM), shale gas, tight gas and hydrate gas. Since 2005, the development of shale gas in the US has become a phenomenon—a revolution in the energy field. This development has had many impacts not only on the US gas market but also on the global gas market.

Unconventional gas production is also growing rapidly in other parts of the world. In 2010, Australia produced only a small amount of coal-bed methane (about 5 billion $m^3$ of gas, in 2015) and became a liquid gas producer from coal-bed methane. Other countries such as China, India and Indonesia also have activities to find and develop non-traditional gas energy sources, including coal-bed methane and shale gas. With the development of shale gas, the proven reserves of natural gas in the United States have increased significantly. Shale gas has helped the USA to overtake Russia to become the largest gas producer in the world since 2009 [34].

The shale gas revolution has led to economic benefits and cost reduction at the state and local levels, individual sectors, and the nation. The exploitation of unconventional gas fields, particularly shale gas, influenced the economic growth of the United States. According to a study in 2014 [35], the macroeconomic impact is relatively limited: around 0.88% growth in the gross domestic product (GDP) between 2007 and 2012. According to the International Monetary Fund report in 2013, the shale gas revolution's macroeconomic impact is between 0.3% and 1% of the US GDP for that year [16]. The shale gas contribution to the American gross domestic product was more than $76.9 billion in 2010; in 2015 it was $118.2 billion and will triple to $230 billion in 2035 [36].

The development of shale gas has helped the US achieve self-sufficiency in energy, improvements in the trade balance and tax revenues. It helped reduce the import of fossil fuels, therefore improving trade balance and simultaneously representing a supplement to the federal budget. In 2012, the sector also generated $62 billion in additional tax revenue for the federal budget, the States, and the concerned municipalities [37].

The development of shale gas in the United States has been the catalyst for the recovery of traditional industries. The products of natural gas-intensive industries can serve as raw materials for the petrochemical industry, fertilizer producers, plastics and other industries that consume a great deal of energy, such as aluminum smelters, steel mills and refineries, etc. The decline of gas prices contributed to the competitiveness enhancement of these sectors in the global competition [15].

### 2.6. Potential of Unconventional Gas

Unconventional gas is considered to play an increasingly important role in securing the global natural gas supply. According to forecasts by the International Energy Organization, non-traditional gas will account for more than 60% of the increase in total gas production from now to 2040.

However, forecasts on natural gas resources still retain a level of uncertainty, especially unconventional natural gas. According to the forecasts in 2017 [17], the renewable resource of traditional natural gas is about 430 trillion $m^3$, allowing about 120 years to be exploited at current production levels (Table 3). For unconventional gas, the forecasted total recoverable shale gas resources are 239 trillion $m^3$, coal-bed methane is 50 trillion $m^3$ and tight gas is 81 trillion $m^3$. The forecast for hydrate gas is 10 times that of shale gas; however, its exploitation technology is complicated. If adding both conventional and non-traditional gas as resources, about 250 years of demand can be satisfied if exploited at their current production rates.

**Table 3.** Forecast of recoverable natural gas resources in the world.

| Region | Traditional Gas (Tcm) | Unconventional Gas | | |
| --- | --- | --- | --- | --- |
| | | Shale Gas (Tcm) | Tight Gas (Tcm) | Coal-Bed Methane (Tcm) |
| Eurasia | 134 | 10 | 10 | 17 |
| Middle East | 103 | 11 | 9 | - |
| Asia Pacific | 44 | 53 | 21 | 21 |
| North America | 50 | 66 | 11 | 7 |
| South America | 28 | 41 | 15 | - |
| Africa | 51 | 40 | 10 | - |
| Europe | 19 | 18 | 5 | 5 |
| Total world | 429 | 239 | 81 | 50 |

Source: [17].

Of the unconventional natural gases, shale gas is the potential gas resource with the largest reserves. Recent studies by scientists have shown that shale gas's potential is huge, its forecast reserves are increasing, and it is widely distributed in many continental countries. This opens many opportunities for its exploitation and use in the future, further contributing to satisfying the demand for natural gas. According to recent publications

by the US Energy Agency and the American Geological Association, the total recoverable resource reserves of shale gas in 46 countries were assessed by the organization to be 7577 Tcf. Shale gas resources are concentrated mainly in China, Argentina, Algeria, and the United States [34,38–41].

### 2.7. The Challenges of Unconventional Gas Development—The Case of Shale Gas

The impact of the production of shale gas on the environment is very strong. The development of shale gas has created significant levels of public concern, and the proportion of its opponents has risen sharply. In this context, this article will analyze the fundamental challenges of shale gas development.

To exploit shale gas, we must use hydraulic fracturing technology. The hydraulic fracturing technology consumes a significant amount of water and chemicals, so it can lead to pollution in the environment throughout the drilling and exploitation process. The production of shale gas consumes a large volume of freshwater. The amount of water needed in the hydraulic fracturing process depends on the type of shale gas and the fracturing operations, such as well depth and length, fracturing fluid properties and fracture job design. In general, 19 million water liters are typically needed per horizontal well in shale gas production [41]. The water consumption will grow with the increase in the number of wells and shale gas production. Certainly, such a large volume of water and a high rate of withdrawals from local surface or groundwater sources has a significant impact on the local water system. Water consumption is particularly important in areas where drought conditions often strictly limit water availability and its use [42,43]. Therefore, the development of shale gas is not recommended in regions or countries that lack water.

### 2.8. The Capacity of Pollution of the Groundwater and Surface Water

The production of shale gas without good practices can contaminate the environment. The chemicals represent from 0.5 to 2% in fluids of hydraulic fracturing; many of them are toxic and carcinogenic. According to an investigative report on the chemicals used in hydraulic fracturing, among the 2500 hydraulic fracturing products, more than 650 are known or possible human carcinogens [36]. Another study identified 632 chemicals used in shale gas operations; more than 75% of the chemicals on the list can affect different organ systems in the body, and more than 50% chemicals indicate effects on the brain and nervous system. These hydraulic fracturing fluids are injected directly into the ground, and they can influence on groundwater sources. In addition, the flowback or "produced" water from fracturing fluid might contaminate the water surface. They may adversely influence human health and the environment quality if they are untreated or directly discharged onto the land or into streams, rivers, and lakes.

### 2.9. Generation of Greenhouse Gases

Shale gas is a type of natural gas that provides cleaner energy than other fossil fuels. However, shale gas contains more than 90% methane ($CH_4$), which may contaminate the air and the water. Methane is a very powerful greenhouse gas compared to carbon dioxide. The effects of shale gas on climate change have become more complex to evaluate and controversial, partly because of uncertainty about the scale of methane leaks. Although it stays only one-tenth of the period compared to carbon dioxide in the atmosphere, methane has a global warming potential 72-fold greater than carbon dioxide when viewed over 20 years and 33-fold greater when viewed over 100 years. Some researchers worry that the expanded production of shale gas could increase methane release as fugitive emissions during the drilling, completion, production, transportation, and the use of natural gas. This is a principal concern because methane is a more potent "greenhouse gas" than $CO_2$, and thus the fugitive emissions in the process of shale gas development may have led to a net increase in greenhouse gas emissions.

### 2.10. The Price of Natural Gas Does Not Cover Operating Costs

The exploitation of shale gas is profitable if the price of natural gas can offset the operational costs. The current price of natural gas in the United States is extremely low—perhaps lower than the actual production cost.

Economists believe that natural gas production marginal cost could certainly reach $4 to $5 per Mbtu [39]. The actual price was approximately $3 per Mbtu in 2012 but was over $4 per Mbtu in 2013 and 2014. This price could be less than the marginal cost of production in the long term with shale gas. In addition, the life of the operation of well shale gas is shorter than that of the production well for conventional gas. Moreover, the life cycle of well shale gas is shorter than the well conventional natural gas, and the production of shale gas declines rapidly after the peak of production (Figure 1). As such, it is necessary to continue the supply of investment capital. At present, some gas producers in the United States are going to reduce their production and their investments in shale gas development activities.

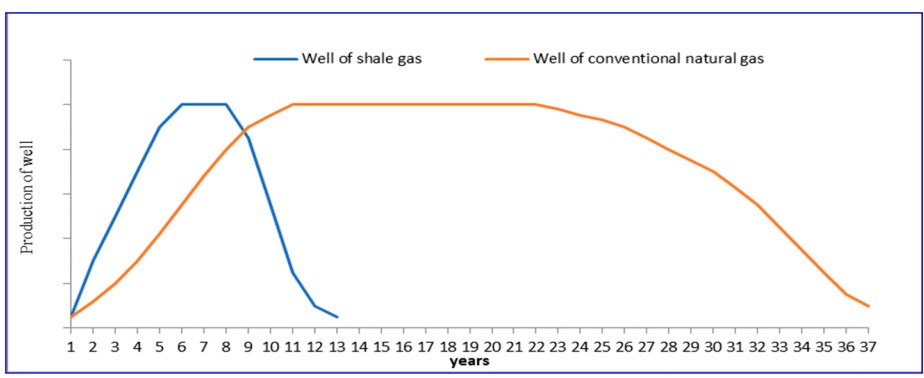

**Figure 1.** The life cycle of a natural gas well. Source: Own study.

### 2.11. The Opposition from the Population

An essential aspect of the development of shale gas and unconventional gas, in general, is the "social license to operate" for activities in this field. According to EIA, the need to build a "social license to operate" was emphasized [39]. The community needs all related information regarding shale gas operations to understand the environmental risks associated with shale gas production. As noted above, shale gas development has created a significant amount of public concern; the percentage of opponents has increased significantly. According to the survey results in Europe, the rate of people who opposed shale gas development in 2013 was over 60% [44]. Another example in Quebec, Canada, shows a rate of opposition at 67%. Therefore, the absence of social acceptability and the hostility of most of the population toward shale gas development will cause significant restrictions in the future.

### 2.12. The Uncertainty of Resource Estimation

The estimates of resources in shale gas are variable and uncertain. There is a lack of serious geological research in the world about the real scope of unconventional reserves. This leads to several different estimates about the shale gas in place and the technically and economically recoverable amount of shale gas in the world [45]. The uncertainty of the estimates will strongly influence the industry's future and the national energy policy. Therefore, the profitability potential of shale gas is still hard to predict. Except for the United States, other countries that are considered to hold significant potential for shale gas resources lack reliable estimates on the technically recoverable resources and economically recoverable resources, which could be a great obstacle in developing shale gases among those countries.

Besides the above challenges, other reasons may constitute obstacles to the development of shale gas, such as the hydraulic fracturing with induced earthquakes, or the

operating processes of shale gas with the destruction of landscapes, the influence on wildlife, and the generation of large amounts of noise.

## 3. Materials and Methods

### 3.1. The Empirical Model

One of the basic and widely used models to explain factors that affect the environment is IPAT model [46]. The name of the IPAT comes from the formulation below:

$$I = P \times A \times T, \tag{1}$$

where I is the pollution or environmental impact, P is the population, A is the capita consumption, or affluence and T is technology. In this paper, we construct our model based on this basic model. In the literature, for the papers that investigated the relationship between any variable and $CO_2$, their independent variables were the gross domestic product (GDP) based on the testing of EKC GDP squared, as well as the population, control variables and the variable that investigated its relationship, such as [47–51]. As such, we did not test the EKC hypothesis in this paper. Our model is simply defined below:

$$CO_2 it = f(GDPit, POPit, NGit), \tag{2}$$

where i indicates each panel country; t indicates the year, simply defined below: $CO_2$ represents the $CO_2$ emissions, GDP represents the GDP and NG represents the per capita natural gas consumption.

### 3.2. Research Methodology

In the empirical application, we have three key steps. The study first tests for cross-sectional dependence by employing the Breusch–Pagan Lagrange multiplier (LM) test, Pesaran scaled LM test, bias-corrected scaled LM test and Pesaran cross-section dependence (CD) test. In the second step, the Pesaran cross-sectionally augmented IPS (Pesaran CIPS) panel unit root is used to check the stationarity of the variables. The third step is the cointegration test section and the Johansen–Fisher panel cointegration test, which is also used in this step. For the fourth step, the panel mean group estimator is used to estimate the long-run and short-run parameters.

#### 3.2.1. Cross-Sectional Dependence Tests

After [52] proposed that countries tend to have strong interdependencies due to the increasing global economic integration, the cross-sectional dependence test began to be used in the literature to provide the decision on the type of panel unit root test to use. We have two categories: first-generation panel unit root tests and second-generation panel unit root tests.

Although we used different tests to test cross sectional-dependence, we have the same null hypothesis that is: no cross-sectional dependence exists within the panel data. The tests we used and their statistics are as follows.

First [53], the Lagrange Multiplier (LM) test statistics are:

$$LM_1 = \sum_{i=1}^{N-1} \sum_{j=i+1}^{N} T_{ij} \hat{\rho}_{ij}^2 \rightarrow \chi^2 \frac{N(N-1)}{2}, \tag{3}$$

Because the Breusch–Pagan LM test statistic is not appropriate for testing in large N settings, [54] proposes the Pesaran LM ($LM_2$ below) and Pesaran CD test, the formulas are below.

$$LM_2 = \sqrt{\left(\frac{1}{N(N-1)}\right)} \sum_{i=1}^{N-1} \sum_{j=i+1}^{N} (T_{ij} \hat{\rho}_{ij}^2 - 1) \rightarrow N(0,1), \tag{4}$$

$$\text{CD} = \sqrt{\left(\frac{2}{\text{N}(\text{N}-1)}\right)} \sum_{i=1}^{\text{N}-1} \sum_{j=i+1}^{\text{N}} \text{T}_{ij}\hat{\rho}_{ij}^2 \rightarrow \text{N}(0,1), \tag{5}$$

Ref. [55] offers a simple asymptotic bias correction for the scaled LM test statistic:

$$\text{LM}_3 = \sqrt{\left(\frac{1}{\text{N}(\text{N}-1)}\right)} \sum_{i=1}^{\text{N}-1} \sum_{j=i+1}^{\text{N}} \left(\text{T}_{ij}\hat{\rho}_{ij}^2 - 1\right) - \frac{\text{N}}{2(\text{T}-1)} \rightarrow \text{N}(0,1), \tag{6}$$

In Equations (3)–(6), $\hat{\rho}_{ij}$ is the correlation coefficient of residuals, and these equations are asymptotically standard normal, with $\text{T}_{ij} \rightarrow \infty$, $\text{N} \rightarrow \infty$ and $\text{N}/\text{T}_{ij} \rightarrow c_{ij} \in (0, \infty)$.

### 3.2.2. Panel Unit Root Test

Stationarity is a time series of features of data that can be tested using unit root tests. If the variables have a unit root, we may have to find spurious regression among them in time series models. Because panel data has time series and cross-sectional units, we have to check the variables' stationarity. Within the panel unit root-testing framework, we have several unit roots tests that are split into two categories: first-generation and second generation. If there is no cross-sectional dependence, the first-generation tests are used; if there is cross-sectional dependence, the second-generation tests are used. The first-generation tests are [56–62]; the second-generation tests are [63–69].

In this paper, we used [68] the cross-sectionally augmented IPS (CIPS) test, which is formulated as follows:

$$\text{CIPS} = \text{N}^{-1} \sum_{i=1}^{\text{N}} t_i(\text{N}, \text{T}), \tag{7}$$

where $t_i(\text{N}, \text{T})$ is the cross-sectionally augmented Dickey–Fuller statistic for the ith cross-section unit.

### 3.2.3. Panel Cointegration Test

Stationarity and cointegration are both time-series features of the data. In the time series econometrics, if the variables are all order-integrated, there will be a long-run relationship, which is called a cointegrated relationship between the variables. One study proposed a cointegration test for the same order=integrated variables in time series econometrics [70]. In the panel data, we have different cointegration frameworks for data with the same integration levels. In this paper, because our variables integration levels are the same, we can use the Fisher-type Johansen panel cointegration test, which is proposed by [59,60] by using the combined test [71]. Other researchers [59,60] adjusted the Johansen test for panel data as follows:

$$\Delta y_{it} = \Pi_{it} + \sum_{j=1}^{k} \Gamma_{it} \Delta y_{it-j} + \varphi_i z_{it} + \varepsilon_{it}, \tag{8}$$

where $y_{it}$ is a $p \times 1$ vector of endogenous variables, p is the number of variables and $\Pi_{it}$ shows the long-run $p \times p$ matrix. The ADF Fisher type test statistics calculated by [59,60] are as follows:

$$p = -2 \sum_{i=2}^{\text{N}} \ln(p_i) \rightarrow \chi_{2\text{N}}^2 \tag{9}$$

The Johansen–Fisher test uses trace and maximum eigenvalue tests. The trace test alternative hypothesis is more than r cointegrated vectors and the maximum eigenvalue test alternative hypothesis is exactly r + 1 cointegrated vectors.

### 3.2.4. Panel Pooled Mean Group (PMG) Model

To estimate the long-run and short=run relationships between variables, the Autoregressive Distributed Lag (ARDL) model can be used. We used the Panel Pooled Mean

Group (PMG) model to estimate the ARDL. The PMG estimator, developed by [61], which estimates the constrain of the long-run coefficients to be identical to the error-correction model, but the long-run coefficients may differ from the error variances. The general formula of the model is as follows:

$$\Delta Y_{it} = \vartheta_i \eta_{it} + \sum_{j=0}^{q-1} \theta'_{ij} \Delta X_{it-j} + \sum_{j=1}^{p-1} \gamma_{ij} \Delta Y_{it-j} + \varepsilon_{it} , \qquad (10)$$

$$\eta_{it} = \delta Y_{it-1} - \beta' X_{it}, \qquad (11)$$

where $Y_{it}$ represents dependent variable $X_{it}$ is ($k \times 1$) vector of explanatory variables, $\theta_{ij}$ denotes coefficients vectors ($k \times 1$), $\gamma_{ij}$ shows the coefficients of lagged variables, $\Delta$ denotes lag operator, $\eta_{it}$ is the error correction term, $\beta$ exhibits long term coefficients and $\vartheta$ denotes adjustment coefficients. If the coefficient of error correction term is negative and significant, the system will return to long-run equilibrium.

This study has novelty because of its examination of the influence of natural gas consumption on $CO_2$ emissions for selected G20 countries, which has not been widely investigated in previous research. Furthermore, this is the first study on the impact of natural gas consumption on $CO_2$ emissions by using the pooled mean group estimator.

## 4. Results of Research

The variables used are $CO_2$, GDP, POP, and NG, which show the $CO_2$ emissions (metric tons per capita), gross fixed capital formation (constant 2015 US\$), population density (people per square km of land area) and natural gas consumption (Exajoules), respectively. Except for NG, the sources of the data are the World Development Indicators database, and the NG collected is from the BP Statistical Review of World Energy. Therefore, the empirical model is expressed as follows:

$$CO_{2it} = \alpha_0 + \delta_{1i}NATG_{it} + \delta_{2i}lgdp_{it} + \delta_{3i}POP_{it} + \varepsilon_{it} , \qquad (12)$$

We used the gross fixed capital formation (GFCF) as capita consumption, like several other researchers [46,72–76]. We used 14 of the G20 countries from 1994 to 2018. The European Union (EU), China, Mexico, Russia, Saudi Arabia, and the USA were removed from the sample. The reasons for removing the countries are because the EU consists of 28 countries, including some G20 countries, and the natural gas consumption of China, Mexico and USA have outlier values among other countries (Figure 2). For Saudi Arabia, we have a lack of data on the gross fixed capital formation variable. Figure 2 shows that China, Mexico, Russia, the USA, and Saudi Arabia are the outliers and very different structures for natural gas consumption. These countries have different economic, industrial and population levels. When we are selecting variables, we try to absorb these big differences, such as using population density instead of population, using $CO_2$ per capita instead of total $CO_2$ emissions, which are affected by the size of the economy.

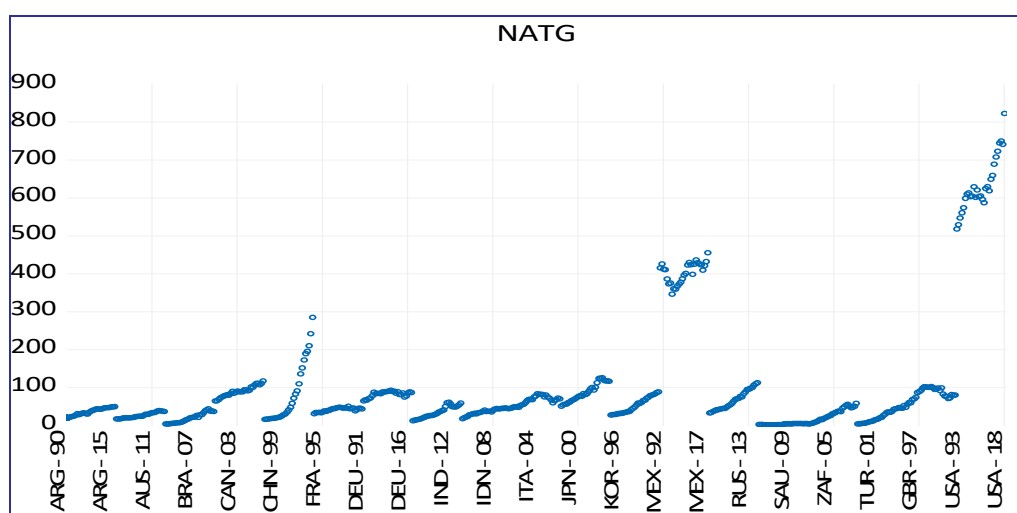

**Figure 2.** Structure of natural gas consumption of the G20 countries. Source: Own study. Notes: Argentina: ARG, Australia: AUS, Brazil: BRA, Canada: CAN, China: CHN, France: FRA, Germany: DEU, India: IND, Indonesia: IDN, Italy: ITA, Japan: JPN, Korea, Rep.: KOR, Mexico: MEX, Russian Federation: RUS, Saudi Arabia: SAU, South Africa: ZAF, Turkey: TUR, United Kingdom: GBR, United States: USA.

There is no consensus about the relationship between these independent variables and $CO_2$ emissions. Some research [51] investigates the effects of stock market growth and renewable energy use on $CO_2$ emissions for G20 countries and estimated three different models for three different samples (full sample, developing economies and developing economies).

The authors used renewable energy consumption in two models; the value of the coefficient of renewable energy is negative in one model and positive in another model. This paper is important for us from the literature based on two features: first, it is on $CO_2$ emissions; second, the cross-section unit of the paper is the G20. The paper did not consider natural gas consumption, but because the expected impact of renewable energy on emissions was similar to the impact of natural gas consumption on emissions, we present its results. The results of [50] for BRIC countries and [77] for 14 Asia-Pacific countries found a negative coefficient for natural gas consumption and a positive coefficient for GDP.

In [77], country-based results have different signs in some countries. Another paper [78] investigated the relationship for sub-Saharan Africa and found that natural gas and GDP positively affected the $CO_2$ emissions. Another paper found a positive relationship between natural gas consumption and $CO_2$ emissions, which investigates Iran and China [79]. Another study we reviewed [80] was a time-series study on Malaysia and found a positive relationship between natural gas consumption and $CO_2$ emissions. One study on the different regions of China and found a positive sign on the natural gas variable in all regions [81].

Table 4 shows the descriptive statistics of the variables. The standard deviations of the variables showed that the selected variables did not have significant variation except for the GDP data. According to descriptive statistics, we decided to take the logarithm of the GDP, and the other variables were used in their level form. Country-level descriptive statistics are presented in Table 4.

**Table 4.** Descriptive statistics of level data and logarithmic data for the 14 countries for the period 1994–2018.

|  | CO$_2$ | GDP | NG | POP | Lgdp [a] |
|---|---|---|---|---|---|
| Mean | 7.352917 | $3.56 \times 10^{11}$ | 53.37571 | 167.2243 | 26.27199 |
| Median | 7.050018 | $3.17 \times 10^{11}$ | 45.65697 | 119.0472 | 26.48341 |
| Maximum | 18.50285 | $1.28 \times 10^{12}$ | 124.7521 | 529.1902 | 27.88171 |
| Minimum | 0.727526 | $2.24 \times 10^{10}$ | 4.739170 | 2.324174 | 23.83196 |
| Std. Dev. | 4.845464 | $2.75 \times 10^{11}$ | 28.35088 | 154.5411 | 0.887738 |
| Obs. | 350 | 350 | 350 | 350 | 350 |

Notes: [a]—logarithmic values are shown using lower letter. Source: Own study.

Table 5 shows the correlation matrix of the variables. Except for the correlation coefficient between POP–CO$_2$ all the correlation coefficients are significant. The significant coefficients are significant in the 1% level of significance at the 1% level. Moreover, all the correlation coefficients are positive. Generally, we can say we have a moderate correlation between variables and the highest between NG and lgdp.

**Table 5.** Correlation matrix.

|  |  | CO$_2$ | lgdp | NG | POP |
|---|---|---|---|---|---|
| CO$_2$ | r | 1.000000 |  |  |  |
|  | p | - |  |  |  |
| lgdp | r | 0.2502 | 1.0000 |  |  |
|  | p | 0.0000 | - |  |  |
| NG | r | 0.4434 | 0.6342 | 1.0000 |  |
|  | p | 0.0000 | 0.0000 | - |  |
| POP | r | −0.0856 | 0.5282 | 0.35033 | 1.0000 |
|  | p | 0.1106 | 0.0000 | 0.0000 | - |

Source: Own study.

Before the model is constructed, the stationarity of the data must be tested because panel data have time-series features. To do this, the test unit root tests and panel unit root tests are used. The panel unit root tests distinguished two generations of the unit root tests on which the first-generation tests relied on the assumption that all cross-sectional units are independent, and for the second generation of panel unit root tests, the presence of cross-sectional dependence [82].

Table 6 shows that all the tests of the cross-sectional dependence gave the same result; the null hypothesis of no cross-sectional dependence is rejected at the 1% level of significance. These results indicate the second generate unit root tests. We used the widely used CIPS test, which is the second-generation test.

**Table 6.** Cross-sectional dependence test.

| Test | Statistic | *p*-Value |
|---|---|---|
| Breusch–Pagan LM | 784.1598 | 0.0000 |
| Pesaran scaled LM | 51.38042 | 0.0000 |
| Bias-corrected scaled LM | 51.08876 | 0.0000 |
| Pesaran CD | 2.3335 | 0.0196 |

Source: Own study.

Based on the CIPS test for the country base, the results for the integration level are different. Some variables are I(0), and some of them are I(1). If we have variables in different integration levels, it is better to choose the ARDL model [61,83,84].

The null hypothesis of the CIPS test is that the variable is homogeneous non-stationary. Table 7 shows that all the variables are I(1); therefore, we decided that they were integrated at the same level and we can use the Johansen–Fisher panel cointegration test to check if there is cointegration.

**Table 7.** CIPS unit root test.

| | Level | | Difference | |
|---|---|---|---|---|
| | Constant | Constant Trend | Constant | Constant Trend |
| $CO_2$ | −0.724 | −2.389 | −4.169 *** | −4.564 *** |
| NG | −2.481 | −1.693 | −3.855 *** | −3.876 *** |
| lgcf | −1.448 | −1.834 | −3.73 *** | −3.841 *** |
| POP | −1.58 | −1.360 | −2.671 *** | −2.985 *** |

Notes: CIPS test developed with the xtcips command of Stata with 5 maximum lags. The critical values at constant are: −2.14, −2.25, −2.45 for 10%, 5%, 1% significance levels, respectively. The critical at values trend are: −2.66, −2.76, −2.96 for 10%, 5%, 1% significance levels, respectively. *** shows rejection of null hypothesis at 1% significance level. Source: Own study.

The null hypothesis of the Fisher test indicated there was no cointegration and we rejected it in all of the hypotheses (Table 8). Thus, we have a cointegration among the variables. We decided to use the ARDL (autoregressive distributed lag) model to investigate the relationship among these variables because it provides the opportunity to see both long-run and short-run relationships. The main feature of cointegrated variables is their responsiveness to any deviation from the long-run equilibrium and based on this feature, the short-run correction from equilibrium can be calculated [85].

**Table 8.** Johansen–Fisher panel cointegration test.

| Hypothesized | Fisher Stat. | | Fisher Stat. | |
|---|---|---|---|---|
| No. of CE(s) | (from Trace Test) | Prob. | (from Max-Eigen Test) | Prob. |
| None | 265.5 | 0.0000 | 223.1 | 0.0000 |
| At most 1 | 92.22 | 0.0000 | 55.41 | 0.0015 |
| At most 2 | 60.80 | 0.0003 | 44.70 | 0.0236 |
| At most 3 | 57.96 | 0.0007 | 57.96 | 0.0007 |

Source: Own study.

According to [59] the panel ARDL model of this paper can be formulated as follows:

$$
\begin{aligned}
\Delta CO_{2it} = {} & \alpha_0 + \sum_{j=1}^{m-1} \delta_{ij} \Delta CO_{2it-1} + \sum_{l=0}^{n-1} \varphi_i \Delta NATG_{it-1} \\
& + \sum_{r=0}^{p-1} \gamma_{i\Delta} lgdp_{it-1} + \sum_{u=0}^{s-1} \theta_i \Delta POP_{it-1} + \beta_1 CO_{2it-1} \\
& + \beta_2 NATG_{it-1} + \beta_3 lgdp_{it-1} + \beta_4 POP_{it-1} + \varepsilon_{it},
\end{aligned}
\tag{13}
$$

Based on Equation (3), the short-run model with error correction term (ECT) is:

$$
\begin{aligned}
\Delta CO_{2it} = {} & \alpha_{CO_2} + \sum_{k=1}^{q} \varphi_{11ik} \Delta CO_{2it-k} + \sum_{k=1}^{q} \varphi_{12ik} \Delta NATG_{it-k} + \sum_{k=1}^{q} \varphi_{13ik} \Delta lgdp_{it-k} \\
& + \sum_{k=1}^{q} \varphi_{14ik} \Delta POP_{it-k} + \lambda_{1i} ECT_{it-1} + \varepsilon_{1it},
\end{aligned}
\tag{14}
$$

$$
\begin{aligned}
\Delta NATG_{it} = {} & \alpha_{NATG} + \sum_{k=1}^{q} \varphi_{21ik} \Delta NATG_{it-k} + \sum_{k=1}^{q} \varphi_{22ik} \Delta CO_{2it-k} + \sum_{k=1}^{q} \varphi_{23ik} \Delta lgdp_{it-k} \\
& + \sum_{k=1}^{q} \varphi_{24ik} \Delta POP_{it-k} + \lambda_{2i} ECT_{it-1} + \varepsilon_{2it},
\end{aligned}
\tag{15}
$$

$$
\begin{aligned}
\Delta lgdp_{it} = {} & \alpha_{lgdp} + \sum_{k=1}^{q} \varphi_{31ik} \Delta lgdp_{it-k} + \sum_{k=1}^{q} \varphi_{32ik} \Delta CO_{2it-k} + \sum_{k=1}^{q} \varphi_{33ik} \Delta NATG_{it-k} \\
& + \sum_{k=1}^{q} \varphi_{34ik} \Delta POP_{it-k} + \lambda_{3i} ECT_{it-1} + \varepsilon_{3it},
\end{aligned}
\tag{16}
$$

$$\Delta POP_{it} = \alpha_{POP} + \sum_{k=1}^{q} \varphi_{41ik}\Delta POP_{it-k} + \sum_{k=1}^{q} \varphi_{41ik}\Delta CO_{2it-k} + \sum_{k=1}^{q} \varphi_{42ik}\Delta NATG_{it-k}$$
$$+ \sum_{k=1}^{q} \varphi_{43ik}\Delta lgdp_{it-k} + \lambda_{4i}ECT_{it-1} + \varepsilon_{4it},$$

(17)

where $\Delta$ is the operator of differentiation, k is the lag length and ECT is the error correction term.

We have two different estimators, which are the mean group (MG) estimator and the pooled mean group (PMG) estimator. The main difference between the two estimators is that MG relies on estimating the N time-series regressions and averaging the coefficients, while the PMG estimator includes a combination of the pooling and averaging of coefficients [59]. To compare these two estimators, we use the Hausman test. The authors of [86] stated that the test of the PMG versus MG was run, and the null hypothesis was that the difference in coefficients was not systematic. When the null hypothesis is rejected, MG is accepted as the best estimator.

The null hypothesis of the Hausman test is the PMG estimator, which is known as the efficient estimator [85]. The Chi-squared test results in Table 9 show that the null hypothesis cannot be rejected, and the pooled mean group estimator is the efficient estimator. The estimated PMG for the panel data is below (Table 10).

**Table 9.** Hausman test; mean group model vs. pooled mean group model.

|  | mg | pmg | Difference | S.E. |
|---|---|---|---|---|
| NG | 0.015068 | 0.015415 | −0.00035 | 0.021759 |
| lgcf | 0.637084 | 1.029484 | −0.3924 | 1.172405 |
| POP | −0.98857 | −0.01379 | −0.97478 | 0.80441 |
| $\chi^2(3) = 2.32$ | | | $p = 0.5083$ | |

Source: Own study.

**Table 10.** Panel short- and long-term coefficients: pooled mean group estimator.

| Long-Run Equation | | | | | |
|---|---|---|---|---|---|
| Variable | NG | lgdp | POP | | |
| Coefficient | 0.1118 *** | 0.0046 *** | 0.0113 *** | | |
| **Short-Run Equation** | | | | | |
| Variable | ECT | Constant | $\Delta$NG | $\Delta$lgdp | $\Delta$pop |
| Coefficient | −0.1622 *** | −3.0746*** | 0.0132 | 1.4231 *** | −3.5403 *** |

Note: *** coefficient statistically significant. Source: Own study.

In the long run, all the variables are significant, and the short-run error correction term (ECT) is negative and significant; these results imply there is a long-run relationship and there is an adjustment from the short-run disequilibrium towards the state of long-run equilibrium. Furthermore, the speed of adjustment coefficient is −0.16. In the short-run, NG did not have a significant effect, but the population had a negative effect and the income (lgdp) of the country had a positive effect on $CO_2$ emissions. At last, if the population increases, the $CO_2$ emissions will increase.

The country-level short-run results showed that countries have idiosyncratic relationships among these variables (Table 11). In some countries, the error correction term is not significant, and this implies there is no short-run adjustment. Moreover, the coefficients of the variables and their significance are different for different countries.

**Table 11.** Pooled mean group estimator coefficients for countries.

|  | Coef. | *p* |  | Coef. | *p* |  | Coef. | *p* |
|---|---|---|---|---|---|---|---|---|
|  | Argentina |  |  | Germany |  |  | Korea |  |
| ECT | −0.1601 | 0.296 | ECT | −0.0587 | 0.246 | ECT | −0.0332 | 0.612 |
| NG | 0.0434 | 0 | NG | 0.0497 | 0 | NG | −0.0333 | 0.317 |
| lgcf | 0.4271 | 0.008 | lgcf | 1.0059 | 0.277 | lgcf | 5.3918 | 0 |
| POP | −2.1387 | 0.073 | POP | −0.0669 | 0.05 | POP | 0.0353 | 0.609 |
| _cons | −3.2561 | 0.313 | _cons | −1.0723 | 0.208 | _cons | −0.3778 | 0.573 |
|  | Australia |  |  | India |  |  | South Africa |  |
| ECT | −0.0171 | 0.759 | ECT | −0.1499 | 0.004 | ECT | −0.4220 | 0.004 |
| NG | −0.0967 | 0.043 | NG | 0.0009 | 0.56 | NG | 0.0119 | 0.465 |
| lgcf | −0.2197 | 0.85 | lgcf | −0.10981 | 0.312 | lgcf | 2.2490 | 0.006 |
| POP | −25.7429 | 0 | POP | −0.0449 | 0.163 | POP | −0.9895 | 0.119 |
| _cons | 0.8862 | 0.149 | _cons | −2.8840 | 0.02 | _cons | −6.9573 | 0.035 |
|  | Brazil |  |  | Indonesia |  |  | Turkey |  |
| ECT | −0.4326 | 0.000 | ECT | −0.1540 | 0.06 | ECT | −0.3594 | 0.024 |
| NG | 0.0138 | 0.000 | NG | −0.0095 | 0.227 | NG | 0.0205 | 0.105 |
| lgcf | −0.0203 | 0.905 | lgcf | −0.0477 | 0.717 | lgcf | 0.1374 | 0.587 |
| POP | 1.6981 | 0.008 | POP | 0.0186 | 0.207 | POP | 0.0834 | 0.565 |
| _cons | −11.3113 | 0.000 | _cons | −3.6518 | 0.051 | _cons | −7.8986 | 0.026 |
|  | Canada |  |  | Italy |  |  | United Kingdom |  |
| ECT | −0.1508 | 0.103 | ECT | 0.0071 | 0.853 | ECT | −0.0014 | 0.983 |
| NG | 0.0609 | 0.000 | NG | 0.03513 | 0 | NG | 0.0156 | 0.112 |
| lgcf | 3.1434 | 0.004 | lgcf | 2.79665 | 0.001 | lgcf | 2.2008 | 0.081 |
| POP | −22.2262 | 0.02 | POP | −0.01156 | 0.732 | POP | −0.1252 | 0.352 |
| _cons | −1.2325 | 0.316 | _cons | 0.02156 | 0.976 | _cons | −0.0546 | 0.955 |
|  | France |  |  | Japan |  |  |  |  |
| ECT | 0.03230 | 0.317 | ECT | −0.3723 | 0.029 |  |  |  |
| NG | 0.0453 | 0.000 | NG | 0.0270 | 0.003 |  |  |  |
| lgcf | 0.6168 | 0.269 | lgcf | 2.3524 | 0.065 |  |  |  |
| POP | −0.2492 | 0.022 | POP | 0.1951 | 0.113 |  |  |  |
| _cons | 0.7625 | 0.305 | _cons | −6.0185 | 0.053 |  |  |  |

Source: Own study.

Besides providing economic benefits and creating jobs, the exploitation and development of unconventional gas projects face many challenges. They can generate impacts on the environment, water, air, and even human health. Therefore, it is necessary to have appropriate energy management models, combining the effective development of these projects with environmental issues and public consensus. Oil and gas companies need to review and change their business models and adjust their development strategies accordingly. This could include debt restructuring, cutting costs, distribution, utilization of profits, avoiding overinvesting in drilling operations to ensure economic efficiency and the creation of attractiveness for investors.

All the abbreviations used in the paper are listed in Table A2 in Appendix A.

## 5. Discussion

An essential aspect of unconventional gas development is the acceptability by local communities for activities in this area because the scale of unconventional gas development is vast, and it can generate environmental and social risks. The success of oil and gas companies in general and unconventional oil and gas companies significantly depends on the well-being of communities. Therefore, the company's activities need to be conducted from a perspective of social and environmental responsibility.

According to the IEA, the need to build a "social license to operate" was highlighted [23]. Therefore, when conducting a project, companies need to consider the local context, study the socio-economic, cultural aspects, and the impacts of the extraction and development activities on the community and environment [87–89]. Companies need to develop principles in oil and gas exploitation activities, transparency of information, especially information related to hydraulic fracturing technology such as principles to ensure people's health, air quality, information on the protection and use of water sources, substances used in the mining process. In other words, oil and gas companies need to

find ways to enhance the positive impacts of economic development and minimize any negative impacts.

Thus, the management model needs to represent all related issues to the following categories in unconventional oil and gas exploitation and development activities such as pre-operational planning; site selection and assessment; site design and construction; flowback water; production operations; and landowner relations. In addition, there is a need for a database to track air quality and emissions; community, cultural and historical factors; human health and safety; and water quality and pollution, etc. Such databases could help stakeholders identify appropriate practices for minimizing impacts to surface resources during planning, design, construction, drilling, operations, reclamation, and monitoring [89].

## 6. Conclusions

Natural gas is a cleaner alternative, and as such, it acts as a "bridge fuel" toward truly clean options. Thus, the demand for this resource will inevitably increase in the future, especially in the context that all countries must work together to prevent global climate change and global warming. Natural gas is an effective medium for short-term choice, a bridging energy medium while humanity is waiting for the transition of energy from traditional to renewable energy sources. Contrary to many previous predictions, with the development of science and technology, technology can allow people to discover and increase global natural gas reserves, especially non-traditional natural gas sources.

Studies have shown a great potential for non-traditional gas, which will significantly add to the reserves and production of natural gas in the world and contribute to the increasing demand for natural gas in the future. Over the past decade, non-traditional gas, especially shale gas, has grown significantly. Their impacts on the economy and the environment are not small. However, reality also shows that we will face many challenges when developing these non-traditional gas resources.

The empirical results showed the coefficient of natural gas consumption is positive in the model. In addition, the effect of the population and income on $CO_2$ emission is positive. Only the effect of natural gas consumption contradicts common expectations, but there are many research papers with the same findings [78–81]. For the panel model and country-by-county cross-section model, the investigated variables have a long-run relationship and generally, there is an adjustment from short-run disequilibrium to the state of long-run equilibrium. The positive coefficient of natural gas consumption may result from its effectiveness comparing increasing fossil fuel use, which is why supporting natural gas consumption aids in the achievement of carbon reduction goals.

The short-run results show that countries have idiosyncratic relationships among the selected variables. In some countries, the error correction term is not significant, and this implies there is no short-run adjustment. Moreover, the coefficients of the variables and their significance are different among the participating countries.

There will be significant barriers that will affect the prospects as well as the role of non-traditional gas in particular—natural gas in general—in the energy transition in the future.

**Author Contributions:** All authors have contributed substantially to this article. Conceptualization, E.T., M.K., L.M.T., T.V.H. and E.U.; methodology, E.T., M.K. and E.U.; software, E.U.; validation, E.U., E.T. and M.K.; formal analysis, E.U. and E.T.; investigation, E.T., M.K., L.M.T., T.V.H. and E.U.; resources, L.M.T., T.V.H. and E.U.; data curation, L.M.T., T.V.H. and E.U.; writing—original draft preparation, E.T., M.K., L.M.T. and T.V.H.; writing—review and editing, M.K. and E.T.; visualization, E.T. and E.U.; supervision, E.T.; project administration, M.K. and E.T.; funding acquisition, E.T. All authors have read and agreed to the published version of the manuscript.

**Funding:** This research received no external funding.

**Conflicts of Interest:** The authors declare no conflict of interest.

## Appendix A

**Table A1.** Fossil fuel emission levels (unit: pounds per billion BTU of energy input).

| Pollutant | Natural Gas | Oil | Coal |
|---|---|---|---|
| Carbon Dioxide | 117,000 | 164,000 | 208,000 |
| Carbon Monoxide | 40 | 33 | 208 |
| Nitrogen Oxides | 92 | 448 | 457 |
| Sulfur Dioxide | 1 | 1 122 | 2 591 |
| Particulates | 7 | 84 | 2 744 |
| Mercury | 0 | 0.007 | 0.016 |

Source: [12].

**Table A2.** List of abbreviations.

| Abbreviation | Meaning |
|---|---|
| ARDL | autoregressive distributed lags |
| BRIC countries | Brazil, Russia, India, and China |
| btu | British thermal unit |
| CADF | Covariate Augmented Dickey-Fuller |
| CBM | coal-bed methane |
| CE | cointegrations |
| CIPS | cross-sectionally augmented Im-Pesaran Shin panel unit root |
| CD | cross-section dependence |
| $CH_4$ | methane |
| $CO_2$ | carbon dioxide |
| Coef. | coefficient |
| _cons | constant |
| ECT | error correction term |
| EIA | U.S. Energy Information Administration |
| EKC | the environmental Kuznets curve |
| Fisher Stat. | Fisher exact test |
| GDP | gross domestic product |
| GFCF | gross fixed capital formation |
| G20 countries | Argentina, Australia, Brazil, Canada, China, France, Germany, India, Indonesia, Italy, Japan, South Korea, Mexico, Russia, Saudi Arabia, South Africa, Turkey, the United Kingdom, the United States, and the European Union |
| IEA | International Energy Agency |
| IMF | International Monetary Fund |
| IPAT | mathematical notation of a formula put forward to describe the impact of human activity on the environment |
| IPS | Im-Pesaran-Shin panel unit root |
| lgdp | Log GDP |
| LM | Lagrange multiplier |
| Mbtu | million British thermal units |
| MG | mean group |
| NG | natural gas |
| Obs. | Observations |
| OECD | Organization for Economic Cooperation and Development |
| p | price |
| PMG | pooled mean group |
| POP | population |
| Prob. | probability |
| S.E. | standard error |
| Std. dev. | standard deviation |
| tcm | trillion cubic meters |
| tcf | trillion cubic feet |
| EU | European Union |
| WEC | The World Energy Council |
| WEO | World Economic Outlook |

Source: Own study.

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
