# Peer review of "Managerial Issues Regarding the Role of Natural Gas in the Transition of Energy and the Impact of Natural Gas Consumption on the GDP of Selected Countries"

_resources, doi:10.3390/resources11050042_

Round 1
Reviewer 1 Report
The manuscript contains all my comments from previous reviews. I don’t have any comments.
Reviewer 2 Report
Thank you for reviewing the paper titled Managerial Issues Regarding the Role of Natural Gas in the Transition of Energy and the Impact of Natural Gas Consumption on the GDP of Selected Countries. It touches interesting and important issues, but must be improved.
The structure of the paper is confusing. Chapter 2 should not contain calculations and tables. Chapters: Methods, Results and Discussion are mixed. Chapter 4 must be divided to 4. Results and 5. Discussion. Results should not contain mathematic formulas - they should be moved to Methods chapter.
Lines 271 - 277 - in case of decrease of number of oil and gas companies it's important to say that even tha tthey play main role in global economy DOI: 10.3390/en14227579. Also COVID caused decrease of energy consumption is and have had significant impact on the world’s energy sector and the shift to renewable energy sources https://doi.org/10.1016/j.enpol.2021.112322
good luck with your paper!
Reviewer 3 Report
The comments of the reviewer have been integrated in the manuscript.
Round 2
Reviewer 2 Report
The authors did not follow my recommendations about the structure of the paper. Chapter 2 should not contain tables and figures, they should be moved to Results Chapter. Results Chapter needs to show results not methods and mathematic formulas.
This manuscript is a resubmission of an earlier submission. The following is a list of the peer review reports and author responses from that submission.
Round 1
Reviewer 1 Report
Manuscript review: Managerial Issues Regarding the Role of Natural Gas: An Energy Management Model and Challenges with Future Unconventional Gas Development (resources-1643151).
The manuscript deals with an overview of the formulation and implementation of the trend of energy transition and use of natural gas in the world.
- The abstract should be redrafted. It is incomprehensible. It will be more understandable if it includes: the sentence of introduction with justification of the presented studies, the purpose of the work, the test methods and the results obtained. The most important results achieved are lacking in the abstract. Please refer to the specific figures obtained from your analysis.
- Why does the introduction not include mechanisms to regulate the energy market, e.g. capacity market mechanisms? These are well-known tools that have been introduced in France, Great Britain, Germany or Poland. I think the absence of this information in connection with the analysis in the manuscript is a major error that will influence the results of the analysis.
- Please delete figures 1-6. Simply provide a link to the website you downloaded from. The form and manner of their presentation is unacceptable in the scientific manuscript. This information is well known.
- Line 120. Remove the table from the introduction. Your description suffices.
- Line 439. Please add a table with all abbreviations and units used in the manuscript.
- Line 542. Novelty must be emphasised. I don’t see her. Executing the statistics does not bring the reader novelty. There are many manuscripts also in the MDPI, which, in addition to statistics for the analysis of the energy market, contain new scientific methods, analyse in detail scenarios of the transformation of the energy market. . . There’s nothing like that here.
- Line 726. Conclusions and discussions are missing in the manuscript. That needs to be reworded.
Reviewer 2 Report
The paper is unreadable in this form. I'm strongly confused about the structure of the paper. Scientific paper needs to have structure: introduction, (theoretical background optionally or in introduction), methods, results, discussion and conclusion.
- Introduction should not contain tables - it's results even yable based on references. the goal of the paper is unclear and doesn't flow from the previous studies. Authors has not stated what new they introduced to existing knowledge.
- Literature review should not contain figures and their analysis. This chapter looks like results not literature review. Figure 6 - current gas prices are available online. Showing prices up to 2017 in case of last few months prices increase is unacceptble. They must be as fresh as it's possible. In this case
- Results and discussion. This chapter should be divided to two separate chapters: Results, Discussion. Mathematic formulas and their explanation needs to be moved to methods chapter. Also I suggest to move to beginning of methods list of studied countries and explanation why authors have choosen them
- Conclusions must contain some words about current gas prices and sources according to Russian war aggression against Ukraine, because it has been strongly changing gas production, selling and policy (especially in Europe) in the world.
According to serious flaws I recommend to reject the paper
Reviewer 3 Report
The paper deals with an interesting and undubtedly hot topic. The method used seems sound and promising, but the general aim and the potential use of results obtained needs to be highlighed and more clearly explained in the text. More specific suggestions are:
- The title is a bit long and does not clearly highlight the aim of the paper;
- The abstract contains some typos and incomplete sentences and must be revised; in addition, it can be rewritten to better highlight the aim and the usefulness of the presented work (who are the main users of this research? who can be the user of the results? do the results confirm a theory or do they indicate a novel trend?);
- The introduction section is interesting and gives a complete picture of the topic and its history; nevertheless, it is too long considering the paper's structure and could be more focused;
- The aim expressed in the introduction is too generic and should be focused not only on what has been done but also on how the outputs of the research can be used and by who;
- The review part is also intereseting but it is lacking a focus on other similar studies, methodologies used and results obtained by other authors;
- Conclusions could be rewritten according to what already stated for abstract and introduction/aim.